# GFAPβ and GFAPδ Isoforms Expression in Mesenchymal Stem Cells, MSCs Differentiated Towards Schwann-like, and Olfactory Ensheathing Cells

**DOI:** 10.3390/cimb47010035

**Published:** 2025-01-09

**Authors:** Nidia Jannette Carrillo González, Gabriela Stefania Reyes Gutierrez, Tania Campos-Ordoñez, Rubén D. Castro-Torres, Carlos Beas Zárate, Graciela Gudiño-Cabrera

**Affiliations:** 1Laboratorio de Desarrollo y Regeneración Neural, Departamento de Biología Celular y Molecular, Centro Universitario de Ciencias Biológicas y Agropecuarias, Universidad de Guadalajara, Zapopan 45220, Jalisco, Mexico; nidia.carrillo@academicos.udg.mx (N.J.C.G.); gabriela.rgutierrez@alumnos.udg.mx (G.S.R.G.); tania.campos@academicos.udg.mx (T.C.-O.); 2Laboratorio de Neurobiotecnología, Departamento de Biología Celular y Molecular, Centro Universitario de Ciencias Biológicas y Agropecuarias, Universidad de Guadalajara, Zapopan 45220, Jalisco, Mexico; dario.castro@academicos.udg.mx (R.D.C.-T.); carlos.beas@academicos.udg.mx (C.B.Z.)

**Keywords:** GFAP isoforms, mesenchymal stem cells, conditioned medium, glial phenotype, Schwann-like cell, cell differentiation

## Abstract

Olfactory ensheathing cells (OECs) and mesenchymal stem cells (MSCs) differentiated towards Schwann-like have plasticity properties. These cells express the Glial fibrillary acidic protein (GFAP), a type of cytoskeletal protein that significantly regulates many cellular functions, including those that promote cellular plasticity needed for regeneration. However, the expression of GFAP isoforms (α, β, and δ) in these cells has not been characterized. We evaluated GFAP isoforms (α, β, and δ) expression by Polymerase Chain Reaction (PCR) assay in three conditions: (1) OECs, (2) cells exposed to OECs-conditioned medium and differentiated to Schwann-like cells (dBM-MSCs), and (3) MSC cell culture from rat bone marrow undifferentiated (uBM-MSCs). First, the characterization phenotyping was verified by morphology and immunocytochemistry, using p75, CD90, and GFAP antibodies. Then, we found the expression of GFAP isoforms (α, β, and δ) in the three conditions; the expression of the GFAPα (10.95%AUC) and GFAPβ (9.17%AUC) isoforms was predominantly in OECs, followed by dBM-MSCs (α: 3.99%AUC, β: 5.66%AUC) and uBM-MSCs (α: 2.47%AUC, β: 2.97%AUC). GFAPδ isoform has a similar expression in the three groups (OEC: 9.21%AUC, dBM-MSCs: 11.10%AUC, uBM-MSCs: 9.21%AUC). These findings suggest that expression of different GFAPδ and GFAPβ isoforms may regulate cellular plasticity properties, potentially contributing to tissue remodeling processes by OECs, dBM-MSCs, and uBM-MSCs.

## 1. Introduction

Glial fibrillary acidic protein (GFAP) is a critical intermediate filament (IF) responsible for the cytoskeleton structure in different cell types that participate in tissue remodeling processes, such as Bergmann glia, stem cells, astrocytes, enteric glia, Schwann cells, and Müller glia [1]. Its expression can also be found in several cell types outside of the nervous system, including fibroblasts, liver stellate cells, chondrocytes, myoepithelial cells, and lymphocytes [2,3]. Posttranslational modifications of GFAP lead to changes in the localization or in the binding of interacting proteins, thereby altering morphology and cellular functions such as cell migration, motility, and mitosis, which are relevant to the role of GFAP in health and disease [4,5,6].

A unique characteristic of GFAP is regulating the pre-RNA transcript by alternative splicing and alternative polyadenylation, which is the main difference compared to other type III intermediate filaments [2]. Indeed, the GFAP gene consists of nine exons and eight introns that can be alternatively spliced to give rise to at least seven splice-isoforms in murines and twelve in humans [2,7]. However, the complete function of GFAP still needs to be fully understood, and the multiple isoforms of GFAP remain relatively unknown. The most abundant isoform, GFAPα, is the predominant isoform in the brain and spinal cord; in contrast, GFAPβ is preferentially expressed in non-myelinated Schwann cells and hepatic stellate cells [8,9,10], cells that are characterized by favoring regeneration processes, where plasticity in the cellular structure is imperative, and GFAPδ is highly expressed by stem cells and neurogenic astrocytes in the subventricular zone, where spherical morphology predominates [11]. In cell culture studies using the immunocytochemistry technique, GFAP protein expression has been reported in bone marrow mesenchymal stem cells (BM-MSCs) that are multipotent precursor cells with the ability to self-renew and exhibit multilineage differentiation capacity [12,13]. These cells also have shown to be able to differentiate toward neural fates and to secrete a broad range of factors able to promote nervous tissue maintenance and repair [14].

As previously observed, olfactory ensheathing cells (OECs), cell culture from rat bone marrow undifferentiated cells (uBM-MSCs) exposed to OECs-conditioned medium (OECs-CM) and differentiated to Schwann-like cells (dBM-MSCs), and uBM-MSCs express GFAP [12,15]. The expression in this group of glial cells is particularly interesting, since it has been described that OECs are widely used in models of regeneration of injuries within the CNS [16], due to its properties that have been shown to contribute to neural regeneration [17].

Therefore, this study aimed to characterize the expression of GFAPα, GFAPβ, and GFAPδ mRNA in alternative sources of cells with a promoter phenotype of regeneration, like BM-MSCs of mesodermal origin [18], the differentiation of BM-MSCs towards OECs phenotype, that is, Schwann-like phenotype, which has been recognized as dBM-MSCs [19]. The differential expression of GFAP isoforms is crucial for tissue and cell-type specificity, significantly influencing development, differentiation, wound-healing process, and various pathological processes [2]. Beyond merely providing structural support to the cell, a specific isoform indicates a functional relationship with a particular cellular process [20]. Therefore, the final objective in this study aimed to characterize the expression of GFAPα, GFAPβ, and GFAPδ mRNA by Polymerase Chain Reaction (PCR) assay in cell culture rat OECs obtained from rat olfactory bulb glial-differentiated MSCs (dBM-MSCs, exposed to OECs-CM) and uBM-MSCs.

## 2. Results

### 2.1. OECs, dBM-MSCs, and uBM-MSCs Phenotypic Characterization

OECs, dBM-MSCs, and uBM-MSCs were analyzed by immunocytochemistry using p75, GFAP, and CD90 antibodies. To confirm the glial phenotype of BM-MSCs after differentiation, we performed immunocytochemistry using markers for Schwann-like cells and OECs such as GFAP and p75. OECs expressed p75 and GFAP proteins, and showed mostly spindle-shaped (Figure 1A, white arrow) and multipolar morphologies (Figure 1A, yellow arrow). After 72 h of differentiation of uBM-MSCs in OECs-CM [21,22], the dMSC-BM also expressed p75 and GFAP markers and exhibited mainly spindle-shape (Figure 1B, white arrow). The uBM-MSCs showed a fibroblastic morphology and expressed the characteristic markers of CD90 and GFAP (Figure 1C, white arrow). Therefore, these data indicate that all groups expressed their characteristic markers and morphological patterns.

### 2.2. OECs, dBM-MSCs, and uBM-MSCs Showed Diffuse GFAP Throughout the Cytoplasm

We analyzed the pattern of GFAP filament organization in OECs, dBM-MSCs, and uBM-MSCs using a 40× objective. We observed that GFAP filaments were diffusely organized in all groups. The OECs showed a diffuse GFAP in mostly spindle-like morphology with long extensions (Figure 2A, white arrows). The dBM-MSCs present the same pattern of GFAP predominately in cell types with spindle-like morphology (Figure 2B, white arrows). The uBM-MSCs showed a diffuse GFAP in cells with fibroblast and flattened morphology (Figure 2C, yellow arrows). GFAP exhibits a uniform, finely granular pattern in all groups, appearing dense and diffusely distributed in the cytoplasm, but excluding the nuclei (Figure 2, asterisk); this pattern was originally reported in cells with active migration [23]. These findings reveal that the GFAP pattern in OECs, dBM-MSCs, and uBM-MSCs may be related to active migration.

### 2.3. OECs, dBM-MSCs, and uBM-MSCs Express GFAPα, GFAPβ, and GFAPδ Isoforms

In the PCR analysis, we examined the GFAP isoforms (α, β, and δ) in OECs, dBM-MSCs, and uBM-MSCs in the PCR analysis. In our analysis, we used the Shapiro–Wilk test to assess the normality of the data. The *p*-values obtained were above 0.05, indicating that the data did not significantly deviate from a normal distribution. Additionally, we employed ANOVA and Tukey’s post hoc tests to determine statistically significant differences between groups. The OECs are characterized by an increased expression of GFAPα (10.96% ± 2.47) with statistically significant differences between groups [F (2, 6) = 7.95, *p* = 0.020], as compared to dBM-MSCs (3.99% ± 1.18; *p* = 0.049) and uBM-MSCs (2.47% ± 0.44; *p* = 0.022). In addition, the expression of GFAPβ was also increased (9.82% ± 0.78) with statistically significant differences between groups [F (2, 6) = 128.1, *p* < 0.0001], as compared to dBM-MSCs (1.03% ± 0.16; *p* < 0.0001) and uBM-MSCs (0.34% ± 0.12; *p* < 0.0001) (Figure 3A–E). However, the three groups expressed a similar amount of GFAPδ (OECs, 7.60% ± 1.94; dBM-MSCs, 8.49% ± 2.43; uBM-MSCs, 7.54% ± 2.24), and no statically significant differences were found between groups (F (2, 6) = 0.057, *p* = 0.94, Figure 3F). These data suggest that OECs, dBM-MSCs, and uBM-MSCs express GFAPα and GFAPβ differently. OECs overexpress these isoforms despite all three cell types exhibiting diffuse GFAP patterns throughout the cytoplasm.

## 3. Discussion

In this study, we characterized the expression of GFAPα, GFAPβ, and GFAPδ mRNA by PCR assay in cell culture rat bone marrow undifferentiated MSCs (uBM-MSCs), glial-differentiated BM-MSCs (dBM-MSCs, exposed to OECs-CM), and OECs obtained from the rat olfactory bulb.

On the one hand, there are novel methods capable of detecting biomarkers in the form of miRNA and mRNA in body fluids including blood and cerebrospinal fluid, as a potential strategy to detect neurodegenerative diseases and inflammatory processes [24,25]. On the other hand, we have GFAP as the main biomarker of astrogliosis for neurodegenerative and neurological diseases; thus, identifying specific GFAP proteoforms could result in testing for specific biomarkers of neurodegenerative diseases, by recognizing the expression of each isoforms that are characteristic of particular cellular processes [25]. For this reason, in this study, we were interested in characterizing the expression of GFAPα, GFAPβ, and GFAPδ mRNA by PCR assay in cell culture of uBM-MSCs, dBM-MSCs, and OECs obtained from the rat olfactory bulb, as a starting point to establish a relationship between GFAP isoforms and the processes involved in the wound healing process.

First, we evaluated the expression of p75, CD90, and GFAP to assess the differentiation of the uBM-MSCs towards the Schwann-like phenotype after 72 h in a medium conditioned by the OECs. The immunocytochemistry results showed that the OECs and dBM-MSCs have predominantly spindle-shaped morphology, with p75+/GFAP+ labeling. uBM-MSCs show a fibroblastoid morphology with positive labeling for p75+/CD90+. Bone marrow MSCs treated with OECs-CM exhibited morphological changes, adopting an elongated spindle phenotype typical of glial cells. OECs-CM promotes the differentiation of MSCs into a glial phenotype, primarily evident through morphological analysis. Additionally, these cells expressed GFAP and p75 proteins after 72 h of treatment, which are also characteristics of the glial phenotype [12,15]. The OECs-CM contains various extrinsic signaling molecules, including growth factors, neurotrophic factors, and cell adhesion molecules [21,26]. These components appear sufficient to induce gene expression associated with cell phenotype, promote neurite outgrowth, and facilitate myelination when applied to stem cells [21,26]. Previously, using the Raman spectroscopy technique, we characterized a significant decrease in the concentration of all major cellular components, including nucleic acids, proteins, and lipids, in differentiated MSCs similar to OECs [15]. Further research is necessary to fully characterize the differentiation of BM-MSCs into this glial phenotype. This characterization should involve qPCR, Western blotting, parallel single-cell RNA sequencing, and in situ hybridization.

In the three groups analyzed, we observed a diffuse pattern of GFAP expression throughout the cytoplasm. PCR analysis shows that OECs, dBM-MSCs, and uBM-MSCs express GFAPα and GFAPβ differently, in which OECs overexpress these isoforms despite all three cell types exhibiting diffuse GFAP patterns throughout the cytoplasm.

OECs, or Schwann-like, are characterized by favoring regeneration processes in the CNS; these cells are distinguished by the expression of p75, a low-affinity receptor for neurotrophins that refers to a characteristic that promotes neural growth, and GFAP, which confers the structure to cellular plasticity [27]. Our study observed that OECs were positive for markers p75 and GFAP, the classic markers used to identify OECs [28,29]. In contrast, uBM-MSCs showed a fibroblastoid morphology with a predominant CD90 expression and the ability to differentiate into adipocytes, osteoblasts, and chondroblasts (Appendix A). Mesenchymal stem cells can be defined by three main criteria, according to the Mesenchymal and Tissue Stem Cell Committee of the International Society for Cellular Therapy. The criteria are adherence to plastic culture ware, expression of surface proteins such as CD90 or CD73, and the ability to differentiate into osteoblasts, adipocytes, and chondroblasts when cultured in a specific differentiating media [30]. Therefore, our findings suggest that murine uBM-MSCs met the international criteria for MSCs. In addition, dBM-MSCs showed a fusiform shape with p75 and GFAP expression, which are markers of phenotypical differentiation to the Schwann-like phenotype. OECs release factors that promote the differentiation of uBM-MSCs towards the Schwann-like phenotype in vitro [12].

The expression of GFAP in OECs, dMSC-BMs, and uBM-MSCs were observed in a punctate and diffuse or sparsity form, which has been associated with a condition that favors cell migration processes [2] and cellular plasticity processes [31]. Stellate-shaped cells, showing paucity of locomotion and relatively rigid postures of processes, contained an abundance of GFAP, which tended to form dense parallel arrays extending into the processes during their development. In contrast, spindle-shaped cells integrate with extending and retracting processes, and active migration also contained an abundance of GFAP, but not organized into parallel arrays [23]. Thus, OECs, uBM-MSCs, and dBM-MSCs exhibit diffuse GFAP patterns throughout the cytoplasm, which may relate to migration or cellular plasticity functions.

GFAP isoforms may regulate cellular plasticity because IFs are specifically expressed in each cell lineage [32]. This strict regulation of the expression of the IF subtype has the potential to give each cell type a unique cytoarchitecture [33,34], which is fundamental in the functioning of the IF network and the physiology of the cell. The regulatory mechanisms that maintain the delicate IF network in optimal conditions are precisely regulated in wound healing processes and favor the regeneration process [35,36,37]. Altered IF dynamics also lead to poor positioning of organelles, compromising energy production and genomic stability, ultimately reducing the ability of the tissue to effectively repair itself [38,39]. In this sense, the alteration of the signals received by the cell can impact essential processes such as cell migration [37,38], proliferation [39], and tissue remodeling [37].

Our data by PCR shows that OECs, dBM-MSCs, and uBM-MSCs express GFAPα and GFAPβ differently. GFAPα is the predominant isoform in the brain and spinal cord [40,41]. This isoform was first described in astrocytes and has been thought of primarily as an astrocyte marker [31]. In our study, OECs showed a higher expression of GFAPα compared to dBM-MSCs and uBM-MSCs. This higher expression in OECs reflects their relationship with their neural source of the olfactory bulbs [27]. On the other hand, dBM-MSCs and uBM-MSCs have a mesenchymal origin; although dBM-MSCs are in transition towards the acquisition of the Schwann-like phenotype, their presence suggests a constant expression of the canonical isoform of GFAP in cells that show cellular plasticity. This report is the first to characterize the expression of three GFAP isoforms in mesenchymal stem cells (MSCs). Until now, the GFAPβ isoform had only been described in Schwann cells. We propose this isoform could be a marker for cells exhibiting cellular plasticity. Understanding this association will enhance our knowledge of expression dynamics in wound repair and provide insights into the functional profiles of cells capable of promoting tissue repair. In this study, OECs demonstrated higher expression levels of GFAPβ compared to dBM-MSCs and uBM-MSCs. Similarly, the GFAPα isoform may be associated with their neural origin. GFAPβ was initially identified in Schwann cells [9], but this isoform is less commonly reported in other cell types. Due to the similarity of immunolabeling with cells of the aldaynoglia (from the Greek aldaynos: to grow) of the CNS, which includes OECs, tanycytes, pituitary cells, and Müller glia, these cells are characterized by significant cellular plasticity and share a series of markers with Schwann cells. This similarity is why they are referred to as Schwann-like cells, sharing the capacity to promote neural regeneration processes, with OECs being the best characterized among them [42]. The GFAP expressed in Schwann cells differs from that expressed by astrocytes and is similar to the GFAP found in fibroblasts and mesenchymal cells [43,44]. This is particularly relevant because the protein product of the GFAPβ isoform is unknown and may play a regulatory role in gene expression. The 5′UTR regions in GFAPβ, but not in GFAPα, can adopt secondary solid structures [45], making them good candidates for the regulatory control of transcription processes [10].

On the other hand, GFAPδ, another isoform highly relevant to the development of the nervous system and expressed inside and outside of the CNS, astrocytes of the subpial, subgranular, and subventricular zone, and precursor cells in general express GFAPδ [40]. This marker distinguishes remaining astrocytes from progenitors of the proliferating subventricular zone [11]. A similar expression of GFAPδ was found by endpoint PCR in the three cell types, OECs, dBM-MSCs, and uBM-MSCs, which may relate to the diffuse punctate GFAP pattern observed by immunocytochemistry. A percentage greater than 10% of GFAPδ is essential to maintain the proportion of GFAPα/GFAPδ, and may be associated with the IF network observed with diffuse spotting [41,46]. GFAPδ is capable of forming heterodimers with other type III IF proteins; thus, depending on the level of expression and the concentration of other IFs present, GFAPδ favors the dispersed state of GFAP, and if the proportion of its concentration increases, it can cause the IF network to collapse (mainly in the perinuclear region) [40,41]. This isoform by itself does not directly influence proliferation or migration, but affects the dynamics of IF networks and the plasticity and motility of cells [11,39,40,47,48].

Furthermore, GFAPδ can modulate intracellular signaling because it is predominantly localized near the nucleus [40] and provides the capacity for remodeling the cytoskeleton, an essential condition in precursor cells [7,49,50]. Considering these observations with the multipotential origin (precursor) of uBM-MSCs and the expression of this isoform, it could be hypothesized that cells that express GFAPα/GFAPδ have characteristics of precursor cells or cells with high plasticity, even if the cell lineage differs from the neural. This indicates the relevance of the characterization of isoforms of GFAP at transcriptional levels to ensure the functionality of cells and favor regeneration processes [51,52,53]. GFAPδ expression in OECs, dBM-MSCs, and uBM-MSCs has not been characterized.

## 4. Materials and Methods

### 4.1. Experimental Animals and Ethics Statement

This work used tissues obtained from Wistar rats (Rattus norvegicus, RRID: RGD_68115). Rodents were housed under optimal environmental conditions and maintained in individual cages in a temperature-controlled room on a 12 h light/dark cycle with ad libitum access to food and water. All experiments were performed following the guidelines of the University of Guadalajara and Official Mexican regulations governing laboratory animal use (NOM-062-ZOO-1999 and NOM-033-ZOO-1995). The experimental procedures were designed to minimize the total number of animals used and the suffering of the animals. Each cell culture employed three rats, for a total of three independent experiments, a total of nine rats (n = 3). The animals were euthanized by CO_2_ asphyxiation and immediately decapitated to extract the tissues for cultures.

### 4.2. Primary Cell Culture of BM-MSCs (uBM-MSCs)

The BM-MSCs isolation culture was performed as previously described [18]. Briefly, the femur and tibia were obtained, the ends were cut at the level of the metaphysis, and the bone marrow was acquired. After passing Hank’s medium without Ca++ and Mg++ (Cat. H2387, Sigma, Saint Louis, MO, USA) through the marrow cavity, it was then centrifuged at 250× *g* for 7 min, the supernatant was removed, and the cells were grown in medium supplemented with 10% FBS and kept under standard culture conditions of 5% CO_2_ and 36.5 °C.

### 4.3. Primary Cell Culture of Olfactory Ensheathing Cells (OECs) and Conditioned Medium Obtention (OECs-CM)

Cultured primary OECs were isolated as described in previous studies [44,54]. Briefly, the olfactory bulbs were dissected to obtain the glomerular nerve fiber layer, then it was mechanically and enzymatically disaggregated with trypsin 2.5 mg (Cat. No. T4174, Sigma–Aldrich, Saint Louis, MO, USA), centrifuged at 250× *g* for 7 min, the supernatant was removed, and the cells were grown in medium supplemented with 10% FBS and kept under standard culture conditions. Once the culture reached cellular confluence, the medium supplemented with 10% FBS was replaced by medium supplemented with B-27 (Cat. 17504-044, Invitrogen, Grand Island, NY, USA), and then their conditioned medium (OECs-CM) was collected. To obtain OECs-CM, purified OECs were cultured in a medium supplemented with B-27.

### 4.4. Differentiation of OECs to Schwann-like Phenotype (dBM-MSCs)

BM-MSCs were differentiated by exchanging 10% FBS culture media for OECs-CM 1:1 with B27 defined medium over 72 h [12].

### 4.5. BM-MSCs Differentiation Analysis

BM-MSCs characteristics were verified by morphology, adipogenic, chondrogenic and osteogenic differentiation potential using several media: Stem MACS AdipoDiff Media (Cat. 130-091-677, Miltenyi Biotec, San Diego, CA, USA), StemMACS ChondroDiff Media (Cat. 130-091-679, Miltenyi Biotec, San Diego, CA, USA), and StemMACS OsteoDiff Media (Cat. 130-091-678, Miltenyi Biotec, San Diego, CA, USA). Further validation was done by observing the distinctive morphology and stain of each expected cell type; osteogenic differentiation was evaluated with alizarin red staining (Cat. A5533, Sigma–Aldrich, Saint Louis, MO, USA); Oil Red O staining (Cat. O9755, Sigma–Aldrich, Saint Louis, MO, USA) was used to examine the ability of adipogenic differentiation; and alcian blue (Cat. 66011, Sigma–Aldrich, Saint Louis, MO, USA) was used to examine the ability of chondrogenic differentiation (Appendix A).

### 4.6. Cell Viability (MTT Assay)

Cell viability was measured using the MTT assay (Cat. M6494, Invitrogen, Grand Island, NY, USA). At the second passage, the uBM-MSCs, dBM-MSCs, and OECs were seeded in 96-well plates, and after three days, 20 μL of MTT (5 mg/mL) was incubated for four hours. Finally, DMSO was added to dissolve the formazan crystals. Absorbance was measured at 590 nm (Appendix A) [12].

### 4.7. Immunocytochemistry

Cells were fixed with 4% paraformaldehyde (Cat. P6148, Sigma–Aldrich, Saint Louis, MO, USA) for 8 min and washed preceding the primary antibody incubation. To identify positive antigens on the cell surface in cells with fibroblast-like morphology (uBM-MSCs), a primary antibody of the cell surface was used: CD90 (Cat. MAB1406, Millipore, RRID: AB_11213488, Temecula, CA, USA). In contrast, to analyze differentiated cells into a glial phenotype, primary antibodies such as mouse monoclonal IgG anti-p75 (Cat. MAB365, Millipore, RRID: AB_2152788, Temecula, CA, USA) and rabbit polyclonal IgG anti-GFAP (Cat. Z0334, DAKO, RRID: AB_10013382, Santa Clara, CA, USA) were used. All of the antibodies were diluted 1:1000 in phosphate-buffered saline (PBS), pH 7.4, with 0.01% bovine serum albumin (Cat. A9418, Sigma–Aldrich, Saint Louis, MO, USA). The incubation period with the anti-p75 was at least 16 h at 4 °C, whereas the incubation period with the GFAP antibody was 1 h at room temperature. Next, the cells were incubated with a corresponding secondary antibody, Alexa-Fluor 488-conjugated anti-mouse IgG antibody (Cat. A21121, Invitrogen, RRID: AB_141514, Grand Island, NY, USA), or Alexa-Fluor 594-conjugated anti-rabbit IgG antibody (Cat. A-11012, Invitrogen, RRID: AB_141359, Grand Island, NY, USA); those secondary antibodies were diluted 1:1000 in PBS and incubated for 45 min at room temperature. Then, cells were covered with Antifade Mounting Medium with DAPI (Cat. H-1200-10, VECTASHIELD, Newark, CA, USA). Finally, the cells were photographed under a 40× objective using an Olympus BX53 fluorescence microscope with a Camera MicroPlublisher 6 and Image Pro imaging software (Version 9.3).

### 4.8. Polymerase Chain Reaction (PCR)

#### 4.8.1. Total RNA Extraction

The cells were obtained by centrifugation, and the culture medium was decanted; the RNA was obtained following the Trizol reagent user guide (Cat. 15596-018, Invitrogen, Grand Island, NY, USA). The RNA pellet was dissolved in 50 µL DEPC water and incubated at 50–60 °C for 10 min. The RNA was then quantified with the Qubit 4.0 fluorometer.

#### 4.8.2. cDNA Synthesis

Reverse transcription was performed following the protocol indicated for M-MLV Reverse Transcriptase (1 μL, 200 U), (Cat. 28025-021, Invitrogen, Grand Island, NY, USA) with Oligo dT (12–18) (0.5 μg/μL) (Cat. 58862, Invitrogen, Grand Island, NY, USA) 1 μL for each reaction, dNTPs (10 mM), (Cat. DNTP10, Sigma-Aldrich, Saint Louis, MO, USA) 1 μL and the necessary volume for 1000 ng of RNA for each reaction, in addition to the buffers supplied by the manufacturer. The cDNA was quantified using fluorescence in Qubit 4.0 and adjusted to 150 ng/μL for PCR.

#### 4.8.3. Nucleic Acids Quantification by Fluorescence

To quantify nucleic acids, total RNA and cDNA (ssDNA), the kits Qubit™ RNA Extended Range (XR) Assay Kit (Cat. Q33223, Invitrogen, Grand Island, NY, USA) and Qubit^®^ ssDNA Assay Kit (Cat. Q10212, Invitrogen, Grand Island, NY, USA) and the Qubit 4.0 were used for their measurement. The samples were prepared as indicated in the user guide (1:200 Qubit buffer/Qubit reagent) for quantification. Each sample was prepared as follows: 195 µL of working solution and 5 µL of sample RNA; and for cDNA (ssDNA), 198 µL of working solution and 2 µL of cDNA were used.

#### 4.8.4. PCR Test

PCR was employed to characterize the expression of GFAP isoforms (α, β, and δ) [10]. In each microtube, the following reaction mix was added: 1X PCR Buffer, 1.5 mM MgCl_2_ (2 mM), 0.2 mM dNTPs, 0.2 μM Forward Primers, 0.2 μM Reverse Primers, injectable H_2_O, Taq Polymerase (5 U/μL) (Cat. EP0402, ThermoScientific, Waltham, MA, USA) 0.5 μL (2.5 U), and ~150 ng of cDNA. The final volume of each tube was 25 μL. The primers used are the same as specified in (Table 1). After being placed in the Techne thermocycler, the following parameters were programmed: an initial denaturation for 5 min at 95 °C, followed by 34 cycles consisting of denaturation for 1 min at 95 °C, hybridization of the primers for 1 min at 62.7 °C, and elongation for 1 min at 72 °C. At the end of the cycles, the sample was kept at 72 °C for 5 min for final elongation.

The amplified products were evaluated on 1.2% agarose gels (Cat. 15510-019, Invitrogen, Grand Island, NY, USA) in 1× TBE. The gels were run at 100 V for around 50 min, and a 100 bp marker (Invitrogen, Cat. 15628019) was used. The products and the marker were stained with GelRed (10,000×), (Cat. 41003, Biotium, Fremont, CA, USA). The fluorescence of the agarose gel was captured using a photodocumenter (ENDURO™ GDS Touch II, Labnet). Subsequently, isoform expression was evaluated using semi-quantitative analysis of the band intensity using the optical densitometry software ImageJ Version 1.54 (NIH, Bethesda, MD, USA) [55].

### 4.9. Statistical Analysis

Data were expressed as the mean ± standard error of the mean (SEM). The normality of data distribution was tested using the Shapiro–Wilk test. ANOVA and Tukey’s post hoc tests determined statistically significant differences between groups. In all cases, the confidence level of statistical significance was 95% (*p* ≤ 0.05). Prism (GraphPad ™ Version 8) was used for data analysis and graph plotting.

## 5. Conclusions

Our results showed significant differences in the expression levels of these isoforms between OECs, dBM-MSCs, and uBM-MSCs. Notably, while GFAPβ was consistently expressed across all three cell types, its co-expression with GFAPδ may correlate with changes in cellular structure that support plasticity and regenerative functions. These results emphasize the role of isoform combinations in shaping cytoskeletal dynamics, potentially driving functional specialization in different cellular contexts. These findings suggest that the expression of GFAPδ and GFAPβ isoforms may participate in cellular plasticity properties, potentially contributing to cellular regeneration processes.

## 6. Limitations of the Study

This descriptive study marks the first exploration of GFAPδ and GFAPβ isoform expression in Schwann-like cells, uBM-MSCs, and dBM-MSCs. This underscores the importance of integrating new approaches, such as in vitro and in vivo assays, to assess the participation of GFAP isoforms in neural repair processes. This article recognizes the methodological limitations that future studies could address. Techniques such as the scratch wound healing assay and migration assays could enhance the current findings [56,57]. These approaches would allow for a more detailed evaluation of GFAP isoform expression and enable their quantitative analysis through PCR, providing deeper insights into their functional roles.

## Figures and Tables

**Figure 1 cimb-47-00035-f001:**
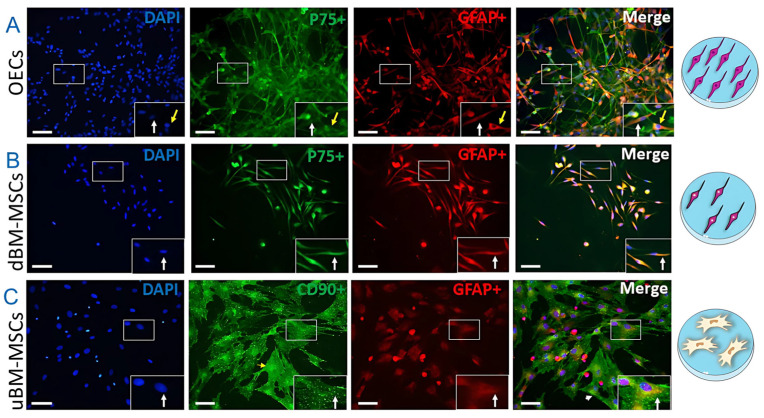
OECs, dBM-MSCs, and uBM-MSCs phenotypic characterization. (**A**) OECs showed elongated spindle-shape and small cell body morphology with bipolar (white arrow) or multipolar (yellow arrow) ramifications and positive expression to p75 and GFAP. (**B**) dBM-MSCs mainly showed elongated spindle shape and small cell body morphology (white arrow) with positive expression to p75 and GFAP. These data suggest that dBM-MSCs had a Schwann-like phenotype after exposure to OECs-CM. (**C**) uBM-MSCs showed a fibroblast-like morphology (white arrow) and positive expression of CD90 and GFAP. OECs: olfactory ensheathing cells, BM-MSC: bone marrow mesenchymal stem cell, dBM-MSC: differentiated BM-MSC, uBM-MSC: undifferentiated BM-MSC. *n* = 3. Bar = 100 µm.

**Figure 2 cimb-47-00035-f002:**
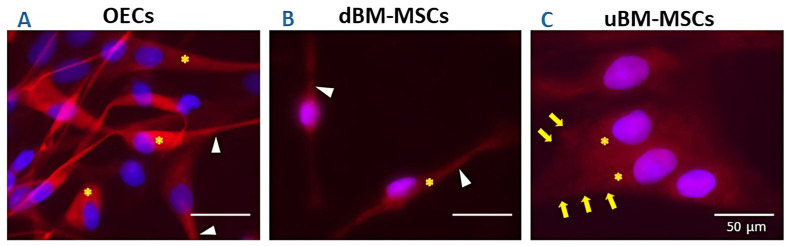
Characterization of GFAP distribution in OECs, dBM-MSCs, and uBM-MSCs. Three groups showed diffuse GFAP throughout the cytoplasm. (**A**) OECs present a mostly spindle-like morphology with long extensions (arrow heads) and diffuse GFAP (asterisk). (**B**) dBM-MSCs expressed GFAP in cell types with spindle-like morphology (arrow heads). (**C**) uBM-MSCs expressed GFAP in cells mostly with morphology similar to fibroblasts (flattened fibroblast) (**yellow arrows**). OECs: olfactory ensheathing cells, BM-MSC: bone marrow mesenchymal stem cell, dBM-MSC: differentiated BM-MSC, uBM-MSC: undifferentiated BM-MSC. *n* = 3. Bar = 50 µm.

**Figure 3 cimb-47-00035-f003:**
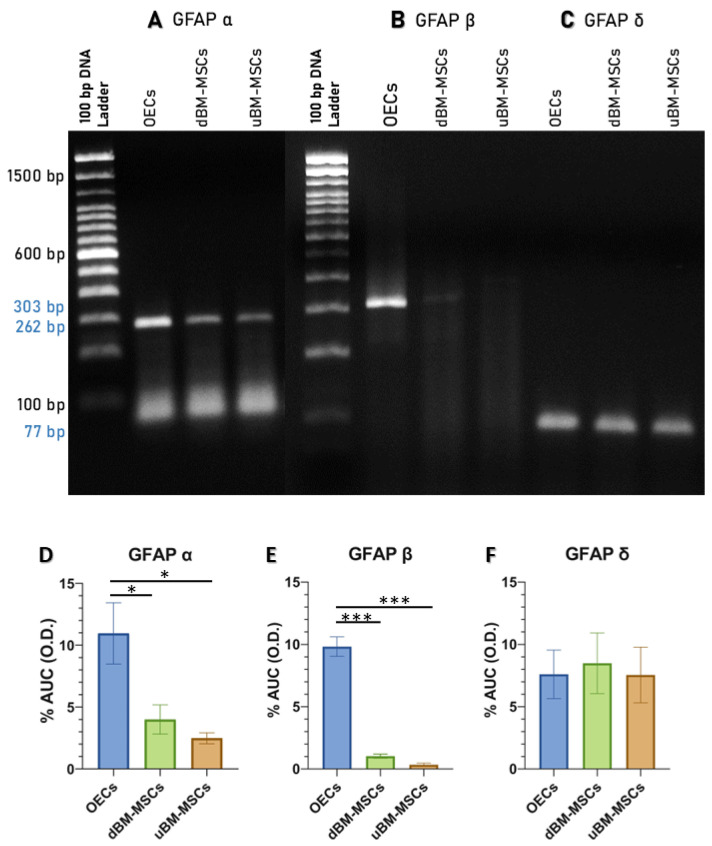
Expression of GFAPα, GFAPβ, and GFAPδ isoforms in OECs, dBM-MSCs, and uBM-MSCs by PCR. (**A**) The expression of the isoforms of GFAPα was observed at 262 bp. (**B**) The expression of the isoforms of GFAP**β** was observed at 303 bp. (**C**) The expression of the isoforms of GFAPδ was observed at 77 bp. Optical densitometry analysis of isoforms, all expressed as a percentage of the area under the curve to GFAPα (**D**); GFAPβ (**E**); GFAPδ (**F**). Data are presented as mean ± SEM. Statistical significance was determined by ANOVA and Tukey’s post hoc tests (*n* = 3) * *p* < 0.05, *** *p* < 0.001. OECs: olfactory ensheathing cells, BM-MSC: bone marrow mesenchymal stem cell, dBM-MSC: differentiated BM-MSC, uBM-MSC: undifferentiated BM-MSC.

**Table 1 cimb-47-00035-t001:** Primers used for the mRNA analysis.

Gene Name	Primer Sequence Forward (+) 5′ → 3′	Primer Sequence Reverse (−) 5′ → 3′	Amplicon Size (bp)
**GFAP α**	GAG ACG TAT CAC CTC TGC AC	GCT GTT CCA GGA AGC GGA CCT	262
**GFAP β**	GAC ATC CCA GGA GCC AGC A	GCT GTT CCA GGA AGC GGA CCT	303
**GFAP δ**	CGT TTT ACT GGT ATG TGG CCC T	TAT GCA CGG CCA ATG TTC CT	77

## Data Availability

The datasets used and analyzed during the current study are available from the corresponding author upon reasonable request.

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
