# Peer review of "GFAPβ and GFAPδ Isoforms Expression in Mesenchymal Stem Cells, MSCs Differentiated Towards Schwann-like, and Olfactory Ensheathing Cells"

_cimb, 2025, doi:10.3390/cimb47010035_

Round 1
Reviewer 1 Report
Comments and Suggestions for Authors
To improve this paper, please carefully consider these points:
- Lines 62-72: "Therefore, this study aimed to characterize the... " Explain here better what the purpose of this paper is and what its final objectives are.
- Lines 205-207: "In our study, OECs showed an increased expression of GFAPα... olfactory bulbs" Discuss this concept more.
- Lines 139-146: "In this study, we characterized the expression of GFAPα, GFAPβ, and GFAPδ mRNA... " Discuss more recent paper ""Furthermore, certain miRNAs can associate with more than one mRNA target, at times within the context of the same signaling pathway" about this topic. Look at recent papers: -- doi: 10.3390/diagnostics13182888 -- doi: 10.1007/s12035-016-0316-2 -- doi: 10.1373/clinchem.2015.239459
- Lines 197-200: "Dysregulation of the IF network impairs essential processes like cell migration... " What do the authors mean with these sentences? Improve this.
- Lines 388-390. Improve this figure legend.
Author Response
Reviewer #1
Reviewer #1 To improve this paper, please carefully consider these points:
Reviewer #1 Lines 62-72: "Therefore, this study aimed to characterize the... " Explain here better what the purpose of this paper is and what its final objectives are.
Response: We appreciate this comment because it points out the need to clarify the purpose and ultimate goals of our article. In response to their suggestion, we have restructured the text to improve its clarity and ensure a smoother flow of information. In addition, we have revised the introduction to provide a more detailed explanation of the main goals and objectives of the study. Specifically, we have emphasized that the main objective of the article is to characterize GFAPα, GFAPβ, and GFAPδ mRNA expression in alternative sources of cells with a regeneration-promoting phenotype, such as BM-MSCs of mesodermal origin. We have also described how our findings contribute to characterizing GFAPα, GFAPβ, and GFAPδ mRNA expression by PCR assays in cell cultures of undifferentiated rat bone marrow MSCs (uBM-MSCs), glial-differentiated MSCs (dBM-MSCs, exposed to OEC-CM), and OECs obtained from the rat olfactory bulb.
Please see: Section of Introduction, Page: 2, Lines: 70-81.
Therefore, this study aimed to characterize the expression of GFAPα, GFAPβ, and GFAPδ mRNA in alternative sources of cells with a promoter phenotype of regeneration, like BM-MSCs of mesodermal origin [18], the differentiation of BM-MSCs towards OECs phenotype, that is, Schwann-like phenotype, which is recognized as dBM-MSCs [19]. The differential expression of GFAP isoforms is crucial for tissue and cell-type specificity, significantly influencing development, differentiation, wound-healing process and various pathological processes [2]. Beyond merely providing structural support to the cell, a specific isoform indicates a functional relationship with a particular cellular process [20]. Therefore, the final objective in this study aimed to characterize the expression of GFAPα, GFAPβ, and GFAPδ mRNA by Polymerase Chain Reaction (PCR) assay in cell culture rat OECs obtained from rat olfactory bulb glial-differentiated MSCs (dBM-MSCs, exposed to OECs-CM) and uBM-MSCs.
Reviewer #1 Lines 205-207: "In our study, OECs showed an increased expression of GFAPα... olfactory bulbs" Discuss this concept more.
Response: We appreciate the comment. In our study, we observed that OECs (Olfactory Ensheathing Cells) exhibited higher expression of GFAPα compared to dBM-MSCs (differentiated bone marrow-derived mesenchymal stem cells) and uBM-MSCs (undifferentiated bone marrow-derived mesenchymal stem cells). This higher expression in OECs reflects their neural origin from the olfactory bulbs, as GFAPα is commonly associated with glial cells in the central nervous system. On the other hand, both dBM-MSCs and uBM-MSCs are of mesenchymal origin. However, it is important to note that dBM-MSCs, which are transitioning towards a Schwann-like phenotype, also express GFAP, in its canonical isoform. This suggests that these cells maintain a level of cellular plasticity, which may contribute to their potential to adapt to different microenvironments, especially in contexts of neural repair or regeneration. This difference in GFAP isoform expression highlights the distinctive properties of these cell types and their potential roles in regenerative medicine.
Please see: Section of Discussion. Page: 7, Lines: 231-237.
In our study, OECs showed a higher expression of GFAPα compared to dBM-MSCs and uBM-MSCs. This higher expression in OECs, reflects their relationship with their neural source of the olfactory bulbs [27]. By contrast, dBM-MSCs and uBM-MSCs have a mesenchymal origin, although dBM-MSCs are in transition towards the acquisition of the Schwann-like phenotype, its presence suggests a constant expression of the canonical isoform of GFAP in cells that show cellular plasticity.
Reviewer #1 Lines 139-146: "In this study, we characterized the expression of GFAPα, GFAPβ, and GFAPδ mRNA... " Discuss more recent paper ""Furthermore, certain miRNAs can associate with more than one mRNA target, at times within the context of the same signaling pathway" about this topic. Look at recent papers: -- doi: 10.3390/diagnostics13182888 -- doi: 10.1007/s12035-016-0316-2 -- doi: 10.1373/clinchem.2015.239459
Response: Thank you for your suggestion and for highlighting the interesting work on the association of miRNAs with multiple mRNA targets. While we recognize the relevance of miRNAs in gene regulation and their potential diagnostic applications, our study is more focused on the characterization of GFAP mRNA isoforms (GFAPα, GFAPβ and GFAPδ). We believe that these isoforms may provide valuable information for the diagnosis of certain pathological conditions. Therefore, our we focused on characterizing the expression of these isoforms and its association with these differentiation conditions, and from the observation we integrate the suggested information.
Please see: Section of Discussion. Page: 5-6, Lines: 157-168.
On the one hand, there are novel methods capable of detecting biomarkers in the form of miRNA and mRNA in body fluids including blood and cerebrospinal fluid, as a potential strategy to detect neurodegenerative diseases and inflammatory processes [24,25] , and on the other hand we have that GFAP is the main biomarker of astrogliosis for neuro-degenerative and neurological diseases, thus, identifying specific GFAP proteoforms could result in testing for specific biomarkers of neurodegenerative diseases, by recog-nizing the expression of each isoforms that are characteristic of particular cellular processes [25] for this reason in this study, we were interested in characterizing the expression of GFAPα, GFAPβ, and GFAPδ mRNA by PCR assay in cell culture of uBM-MSCs), dBM-MSCs and OECs obtained from the rat olfactory bulb, as a starting point to establish a relationship between GFAP isoforms and the processes involved in the wound healing process.
- Wang, W. X., Fardo, D. W., Jicha, G. A., & Nelson, P. T. (2017). A Customized Quantitative PCR MicroRNA Panel Provides a Technically Robust Context for Studying Neurodegenerative Disease Biomarkers and Indicates a High Correlation Between Cerebrospinal Fluid and Choroid Plexus MicroRNA Expression. Molecular neurobiology, 54(10), 8191–8202. https://doi.org/10.1007/s12035-016-0316-2.
- Gogishvili, D., Honey, M. I. J., Verberk, I. M. W., Vermunt, L., Hol, E. M., Teunissen, C. E., & Abeln, S. (2025). The GFAP proteoform puzzle: How to advance GFAP as a fluid biomarker in neurological diseases. Journal of neurochemistry, 169(1), e16226. https://doi.org/10.1111/jnc.16226.
Reviewer #1 Lines 197-200: "Dysregulation of the IF network impairs essential processes like cell migration... " What do the authors mean with these sentences? Improve this.
Response: We appreciate your comment. To clarify our statement, we include the explanation that the intermediate filament (IF) network is strictly regulated in various key cellular processes that are essential for normal tissue development and functioning, and in this sense, the alteration of the dynamics of these filaments can harm cellular integrity and affect several signaling pathways that regulate these processes.
Please see: Section of Discussion. Page: 7, Lines: 222-228.
The regulatory mechanisms that maintain the delicate IF network in optimal conditions are precisely regulated in wound healing processes and favor the regeneration process [35–37]. Altered IF dynamics also lead to poor positioning of organelles, compromising energy production and genomic stability, ultimately reducing the ability of the tissue to effectively repair itself [38,39]. In this sense, the alteration of the signals received by the cell can impact essential processes such as cell migration [37,38], proliferation [39] and tissue remodeling [37].
Reviewer #1 Lines 388-390. Improve this figure legend.
Response: Thank you for your suggestion. We have extended the explanation in the figure legend to provide a clearer and more detailed description of what the figure represents.
Please see: Section of Supplementary Materials. Page: 12, Lines: 430-433.
Supplementary material Figure 2. BMSCs differentiation was identified by morphology of (A) Adipogenic differentiation was confirmed by lipid droplets, (B) Chondrogenic differentiation was confirmed by Alcian Blue staining and (C) Osteogenic differentiation was evaluated with alizarin red staining.
Reviewer 2 Report
Comments and Suggestions for Authors
Psper titled (GFAPβ and GFAPδ isoforms expression in mesenchymal stem cells, MSCs differentiated towards Schwann-like and Olfactory Ensheathing Cells) by Nidia Jannette Carrillo-González et al. is a molecular study that evaluated GFAP isoforms (α, β and δ) expression by PCR assay in 1) cell culture from rat bone marrow undifferentiated (uBM-MSCs),cells exposed to olfactory ensheathing cells-conditioned medium and differentiated to Schwann-like cells (dBM-MSCs), and olfactory ensheathing cells obtained from the olfactory bulb. Authors also Characterized the phenotyping verified by morphology and immunocytochemistry using p75, CD90, and GFAP antibodies .
This is a useful study and well written with a clear straightforward conclusion. I have these recommendations for improving the final shape of the manuscript .
1- Abstract must be amended by some numerical values for key findings from the study .
2- key words: please add (the types of cells used in the study as key words) .
3- Introduction: is brief and did not introduce the cell types tested in the study well, did not explore the rational or novelty of the study .
4- AIm of the study should be clear and clarify what was the aim and how authors achieved it .
5- Methods section in general is too brief and lacks references MUST be extensively revised .
6- How many rats ? how many total animals used ?
7- In methods , Mention in details the housing conditions and how authors were keen to reduce animal suffering .
8- Table 1 should not be an image, please provide an editable form
9- Animal details and housing should be written in details (cage type, number per cage, food, dark light cycles, how minmized animal suffering...etc) .
10- Methods in general lacks references at many occasions .
11- In each illustration mention the type of the presented data & the statistical test applied for analysis .
12- Ensure every abbreviation is explained at the first appearnace in abstract & then in the body text .
13- Authors should give the source of chemicals, kits and antibodies completely and consistently (code, company, town, state and country) & version for software .
14- Authors checked the normality of distribution of the results by Shapiro-Wilk test before deciding to choose certain ANOVA .
15- Authors should confirm in methods that "every possible comparison between the study groups was considered" and apply this in results .
16- Use appropriate abbreviations for minutes, seconds...etc .
17- Every abbreviation in figures should be explained in the figure legend to be self explanatory & stands alone .
18- Mention "n" in each illustration individually .
19- Please mention what is the clinical potential of this study and how it can be useful for treating diseases .
20- Please mention the limitations of this study and future directions after which .
Author Response
Reviewer #2
We appreciate the comment of reviewer #2 who stated “Paper titled (GFAPβ and GFAPδ isoforms expression in mesenchymal stem cells, MSCs differentiated towards Schwann-like and Olfactory Ensheathing Cells) by Nidia Jannette Carrillo-González et al. is a molecular study that evaluated GFAP isoforms (α, β and δ) expression by PCR assay in 1) cell culture from rat bone marrow undifferentiated (uBM-MSCs),cells exposed to olfactory ensheathing cells-conditioned medium and differentiated to Schwann-like cells (dBM-MSCs), and olfactory ensheathing cells obtained from the olfactory bulb. Authors also Characterized the phenotyping verified by morphology and immunocytochemistry using p75, CD90, and GFAP antibodies.
This is a useful study and well written with a clear straightforward conclusion. I have these recommendations for improving the final shape of the manuscript.”
Reviewer #2 1- Abstract must be amended by some numerical values for key findings from the study.
Response: Thank you for your valuable comments. We have revised the abstract to include sequential values ​​for the key findings of the study. In addition, we have revised the logical order in which the cell types analyzed are numbered to improve clarity and consistency.
Please see: Section of Abstract. Page: 1, Lines: 21-31.
1) OECs, 2) cells exposed to OECs-conditioned medium and differentiated to Schwann-like cells (dBM-MSCs), and 3) cell culture from rat bone marrow undifferentiated (uBM-MSCs). First, the characterization phenotyping was verified by morphology and immunocytochemistry, using p75, CD90, and GFAP antibodies. Then, we found the expression of GFAP isoforms (α, β and δ) in the three conditions; the expression of the GFAPα (10.95 %AUC) and GFAPβ (9.17 %AUC) isoforms was predominantly in OECs, followed by dBM-MSCs (α: 3.99%AUC, β: 5.66 %AUC) and uBM-MSCs (α: 2.47%AUC, β: 2.97%AUC). GFAPδ isoform has a similar expression in the three groups (OEC: 9.21 %AUC, dBM-MSCs: 11.10 %AUC, uBM-MSCs: 9.21 %AUC). Conclusion: These findings suggest that expression of different GFAPδ and GFAPβ isoforms may regulate cellular plasticity properties, potentially contributing to tissue remodeling processes by OECs, dBM-MSCs, and uBM-MSCs.
Reviewer #2 2- key words: please add (the types of cells used in the study as key words) .
Response: Thank you for your suggestion regarding the keywords. In response, we have decided to include "Schwann-like" and "cell differentiation" as keywords, as they are particularly relevant and significant to the study. We appreciate your feedback, which has helped us enhance the focus and accessibility of our research.
Please see: Section of Keywords. Page: 1, Lines: 34.
Schwann-like cell 5, Cell differentiation 6.
Reviewer #2 3- Introduction: is brief and did not introduce the cell types tested in the study well, did not explore the rational or novelty of the study.
Response: Thank you for your insightful comment. We have revised the introduction to provide a more comprehensive overview of the cell types tested in the study. Specifically, we have integrated the importance of studying GFAP isoforms in olfactory ensheathing cells (OECs) and bone marrow-derived mesenchymal stem cells (BM-MSCs), highlighting their properties that promote tissue regeneration processes. This addition emphasizes the rationale and novelty of our research, aligning it with the study's objectives.
Please see: Section of Introduction. Page: 1-2,
Lines: 36-39.
Glial fibrillary acidic protein (GFAP) is a critical intermediate filament (IF) responsible for the cytoskeleton structure in different cell types that participate in tissue remodeling processes, such as Bergmann glia, stem cells, astrocytes, enteric glia, Schwann cells, and Müller glia [1].
Lines: 56-61.
In cell culture studies using the immunocytochemistry technique, GFAP protein ex-pression has been reported in bone marrow mesenchymal stem cells (BM-MSCs) that are multipotent precursor cells with the ability to self-renew and exhibit multilineage differentiation capacity [12,13], these cells also shown to be able to differentiate toward neural fates and to secrete a broad range of factors able to promote nervous tissue maintenance and repair [14].
Lines: 66-69.
The expression in this group of glial cells is particularly interesting since it has been described that OECs cells are widely used in models of regeneration of injuries within the CNS [16], due to its properties that have been shown to contribute to neural regeneration [17].
- Volkman, R., & Offen, D. (2017). Concise Review: Mesenchymal Stem Cells in Neurodegenerative Diseases. Stem cells (Dayton, Ohio), 35(8), 1867–1880. https://doi.org/10.1002/stem.2651
- Yao, R., Murtaza, M., Velasquez, J. T., Todorovic, M., Rayfield, A., Ekberg, J., Barton, M., & St John, J. (2018). Olfactory Ensheathing Cells for Spinal Cord Injury: Sniffing Out the Issues. Cell transplantation, 27(6), 879–889. https://doi.org/10.1177/0963689718779353
Reviewer #2 4- Aim of the study should be clear and clarify what was the aim and how authors achieved it .
Response: We appreciate your valuable comments. We have revised the manuscript to emphasize that the main objective of the article was to characterize GFAPα, GFAPβ, and GFAPδ mRNA expression in alternative cell sources with a regeneration-promoting phenotype, such as BM-MSCs of mesodermal origin. We have also described how our findings contribute to characterizing GFAPα, GFAPβ, and GFAPδ mRNA expression by PCR assays in cell cultures of undifferentiated rat bone marrow MSCs (uBM-MSCs), glial differentiated MSCs (dBM-MSCs, exposed to OECs-CM), and OECs obtained from the rat olfactory bulb. We believe that this adjustment better explains how this objective was achieved. We appreciate your comment, which has helped us improve the focus and clarity of our work.
Please see: Section of Introduction. Page: 2, Lines: 70-81.
Therefore, this study aimed to characterize the expression of GFAPα, GFAPβ, and GFAPδ mRNA in alternative sources of cells with a promoter phenotype of regeneration, like BM-MSCs of mesodermal origin [18], the differentiation of BM-MSCs towards OECs phenotype, that is, Schwann-like phenotype, which has been recognized as dBM-MSCs [19]. The differential expression of GFAP isoforms is crucial for tissue and cell-type specificity, significantly influencing development, differentiation, wound-healing process and various pathological processes [2]. Beyond merely providing structural support to the cell, a specific isoform indicates a functional relationship with a particular cellular process [20]. Therefore, the final objective in this study aimed to characterize the expression of GFAPα, GFAPβ, and GFAPδ mRNA by Polymerase Chain Reaction (PCR) assay in cell culture rat OECs obtained from rat olfactory bulb glial-differentiated MSCs (dBM-MSCs, exposed to OECs-CM) and uBM-MSCs.
Reviewer #2 5- Methods section in general is too brief and lacks references MUST be extensively revised.
And
Reviewer #2 10- Methods in general lacks references at many occasions.
Response: Thank you for your important comment. We have extensively revised the Materials and Methods section to provide a more detailed description of the procedures and have taken time to review that most sections include adequate references to support the methodologies used. These changes ensure the reproducibility of the study.
Please see: Section of Materials and Methods. Page: 8-11
Line: 295
The BM-MSCs isolation culture was performed as previously described [18].
Line: 304
Cultured primary OECs were isolated as described in previous studies [44,54]
Lines: 315-316
BM-MSCs were differentiated by exchanging 10% FBS culture media for OECs-CM 1:1 with B27 defined medium throughout 72 h [12].
Lines: 330-334
Cell viability was measured using the MTT assay (Cat. M6494, Invitrogen, Grand Island, NY, USA). At the second passage, the uBM-MSCs, dBM-MSCs, and OECs were seeded in 96-well plates, and after three days, 20 μl of MTT (5 mg/ml) was incubated for four hours. Finally, DMSO was added to dissolve the formazan crystals. Absorbance was measured at 590 nm (Supplementary Figure 2) [12].
Line: 379
PCR was employed to characterize the expression of GFAP isoforms (α, β, and δ) [10].
Lines: 395-396
Subsequently, isoform expression was evaluated using semi-quantitative analysis of the band intensity using the optical densitometry software ImageJ (NIH) [55].
Reviewer #2 6- How many rats ? how many total animals used ?
Response: Thank you for your kind suggestion. In response, we have clarified in the text that a total of 9 rats were used in the study, with 3 rats utilized for each culture preparation. Additionally, we specified that three independent tests were conducted, which is why the sample size is presented as n=3.
Please see: Section 4.1 Experimental animals and ethics statement. Page: 8, Lines: 289-292.
The experimental procedures were designed to minimize the total number animals used and the suffering of the animals. Each cell culture employed three rats, for a total of three independent experiments, a total of nine rats (n=3).
Reviewer #2 7- In methods , Mention in details the housing conditions and how authors were keen to reduce animal suffering .
and
Reviewer #2 9- Animal details and housing should be written in details (cage type, number per cage, food, dark light cycles, how minmized animal suffering...etc).
Response: We thank Reviewer #2 for raising this important concern. We have added further details to the Methods section regarding animal housing and euthanasia. Our commitment to the 3Rs (Replacement, Reduction, and Refinement) guided all aspects of this study, including the choice of euthanasia method and the provision of appropriate housing and care. The animals were housed in solid material cages under controlled conditions, in compliance with the Mexican regulation NOM-062-ZOO-1999 and NOM-033-ZOO-1995, which specifies the following parameters: 45–60% humidity, 22–26°C temperature, and a 12-hour light-dark cycle. To minimize suffering during euthanasia, a gradual fill method for CO2 delivery was employed to reduce distress, followed by immediate decapitation to ensure rapid and irreversible termination of brain function and allow for optimal tissue preservation for subsequent cultures. This combination of methods is consistent with established best practices for rodent euthanasia. We appreciate your suggestion, which has helped us provide greater transparency and compliance with ethical standards.
Please see: section 4.1 Experimental animals and ethics statement. Page 8. Lines: 287-293.
All experiments were performed following the guidelines of the University of Guadalajara and Official Mexican regulations governing laboratory animal use (NOM-062-ZOO-1999 and NOM-033-ZOO-1995). The experimental procedures were designed to minimize the total number animals used and the suffering of the animals. Each cell culture employed three rats, for a total of three independent experiments, a total of nine rats (n=3). The animals were euthanized by CO2 asphyxiation and immediately decapitated to extract the tissues for cultures.
Reviewer #2 8- Table 1 should not be an image, please provide an editable form
Response: We thank Reviewer #2 for this suggestion. We have now replaced Table 1 with an editable format in the revised submission.
Please see: Section 4.8.4 PCR test. Page 11. Line: 389.
Table 1 Primers used for the mRNA analysis |
|||
Gene Name |
Primer sequence Forward (+) 5'->3’ |
Primer sequence Reverse (-) 5'->3’ |
Amplicon Size (bp) |
GFAP α |
GAG ACG TAT CAC CTC TGC AC |
GCT GTT CCA GGA AGC GGA CCT |
262 |
GFAP β |
GAC ATC CCA GGA GCC AGC A |
GCT GTT CCA GGA AGC GGA CCT |
303 |
GFAP δ |
CGT TTT ACT GGT ATG TGG CCC T |
TAT GCA CGG CCA ATG TTC CT |
77 |
Reviewer #2 11- In each illustration mention the type of the presented data & the statistical test applied for analysis .
Response: We appreciate Reviewer #2's feedback. We have now revised all figure captions to clearly indicate the type of data, and the specific statistical test used for analysis. We believe this provides the necessary clarity for the reader.
Please see: Section of Materials and Methods. Figure 3. & Supplementary Materials. Supplementary material Figure 1.
Page 5. Lines: 144-151.
Figure 3. Expression of GFAPα, GFAPβ, and GFAPδ isoforms in OECs, dBM-MSCs, and uBM-MSCs by PCR. (A) The expression of the isoforms of GFAPα was observed at 262 bp. (B) The expression of the isoforms of GFAPδ was observed at 77 bp. (D) Optical densi-tometry analysis of isoforms, all expressed as a percentage of the area under the curve to GFAPα (D); GFAPβ (E); GFAPδ (F). Data is presented as mean ± SEM. Statistical significance was determined by ANOVA and Tukey’s post hoc tests (n=3). OECs: olfactory ensheathing cells, BM-MSC: Bone marrow-mesenchymal stem cell, dBM-MSC: differen-tiated BM-MSC, uBM-MSC: undifferentiated BM-MSC.
Page 12. Line: 427.
Supplementary material Figure 1. Cell viability by MTT assay. Data is presented as mean ± SD (n=3).
Reviewer #2 12- Ensure every abbreviation is explained at the first appearance in abstract & then in the body text .
Response: Thank you for your helpful observation. We reviewed the manuscript and identified the omission of explaining one abbreviation in the introduction. The necessary correction has now been made to ensure that every abbreviation is clearly explained upon its first appearance in both the abstract and the body text. We appreciate your feedback, which has allowed us to improve the clarity of our work.
Please see: Section of Introduction, Page 2, line: 63-66.
As previously observed, olfactory ensheathing cells (OECs), cell culture from rat bone marrow undifferentiated cells (uBM-MSCs) exposed to OECs-conditioned medium (OECs-CM) and differentiated to Schwann-like cells (dBM-MSCs), and 3) uBM-MSCs ex-press GFAP [12,15].
Please see: Section of Abstract, Page 1, line: 16-23.
Olfactory Ensheathing Cells (OECs), Mesenchymal stem cells (MSCs) and MSCs differentiated towards Schwann-like have plasticity properties. These cells express the Glial fibrillary acidic protein (GFAP), a type of cytoskeletal protein that significantly regulates many cellular functions, including those that promote cellular plasticity needed for regeneration. However, the expression of GFAP isoforms (α, β, and δ) in these cells has not been characterized. We evaluated GFAP isoforms (α, β and δ) expression by Polymerase Chain Reaction (PCR) assay in three conditions: 1) OECs, 2) cells exposed to OECs-conditioned medium and differentiated to Schwann-like cells (dBM-MSCs), and 3) MSCs cell culture from rat bone marrow undifferentiated (uBM-MSCs).
Reviewer #2 13- Authors should give the source of chemicals, kits and antibodies completely and consistently (code, company, town, state and country) & version for software .
Response: We appreciate your valuable feedback. We have revised the manuscript to include complete and consistent information for all chemicals, kits, and antibodies, specifying the code, company, city, state, and country. In addition, the version of the software used has been added. We appreciate your feedback, which has helped us ensure greater accuracy and clarity in our documentation.
Please see: Section of Materials and Methods. Pages: 8-11
Lines: 297-298.
After passing Hank's medium without Ca++ and Mg++ (Cat. H2387, Sigma, Saint Louis, MO, USA)
Lines: 306-307.
trypsin 2.5 mg (Cat. No. T4174, Sigma–Aldrich, Saint Louis, MO, USA)
Lines: 310-311.
B-27 (Cat. 17504-044, Invitrogen, Grand Island, NY, USA)
Lines: 320-322.
Stem MACS AdipoDiff Media (Cat. 130-091-677, Miltenyi Biotec, San Diego, CA, USA), StemMACS ChondroDiff Media (Cat. 130-091-679, Miltenyi Biotec, San Diego, CA, USA) and StemMACS OsteoDiff Media (Cat. 130-091-678, Miltenyi Biotec, San Diego, CA, USA).
Lines: 324-327.
alizarin red staining (Cat. A5533, Sigma–Aldrich, Saint Louis, MO, USA); Oil Red O staining (Cat. O9755, Sigma–Aldrich, Saint Louis, MO, USA) was used to examine the ability of adipogenic differentiation; and alcian blue (Cat. 66011, Sigma–Aldrich, Saint Louis, MO, USA).
Lines: 330-331.
MTT assay (Cat. M6494, Invitrogen, Grand Island, NY, USA).
Lines: 336-337.
4% paraformaldehyde (Cat. P6148, Sigma–Aldrich, Saint Louis, MO, USA)
Lines: 339-345
CD90 (Cat. MAB1406, Millipore, RRID: AB_11213488, Temecula, CA, USA). In contrast, to analyze differentiated cells into a glial phenotype, primary antibodies such as mouse monoclonal IgG anti-p75 (Cat. MAB365, Millipore, RRID: AB_2152788, Temecula, CA, USA) and rabbit polyclonal IgG anti-GFAP (Cat. Z0334, DAKO, RRID: AB_10013382, Santa Clara, CA, USA) were used. All the antibodies were diluted 1:1000 in phosphate-buffered saline (PBS), pH 7.4, with 0.01% bovine serum albumin (Cat. A9418, Sigma–Aldrich, Saint Louis, MO, USA).
Lines: 348-351
Alexa-Fluor 488-conjugated anti-mouse IgG antibody (Cat. A21121, Invitrogen, RRID: AB_141514, Grand Island, NY, USA), or Alexa-Fluor 594-conjugated anti-rabbit IgG antibody (Cat. A-11012, Invitrogen, RRID: AB_141359, Grand Island, NY, USA);
Lines: 353-355.
Antifade Mounting Medium with DAPI (Cat. H-1200-10, VECTASHIELD, Newark, CA, USA). Finally, the cells were photographed under 40× objective using an Olympus BX53 fluorescence microscope with a Camera MicroPlublisher 6 and Image Pro imaging software (Version 9.3).
Lines: 359-360.
Trizol reagent user guide (Cat. 15596-018, Invitrogen, Grand Island, NY, USA).
Lines: 363-366.
M-MLV Reverse Transcriptase (1 μL, 200 U), (Cat. 28025-021, Invitrogen, Grand Island, NY, USA) with Oligo dT (12-18) (0.5 μg/μL) (Cat. 58862, Invitrogen, Grand Island, NY, USA) 1 μL for each reaction, dNTPs (10 mM); (Cat. DNTP10, Sigma-Aldrich, Saint Louis, MO, USA)
Lines: 372-374.
the kits Qubit™ RNA Extended Range (XR) Assay Kit (Cat. Q33223, Invitrogen, Grand Island, NY, USA) and Qubit® ssDNA Assay Kit (Cat. Q10212, Invitrogen, Grand Island, NY, USA)
Lines: 382.
Taq Polymerase (5U /μL) (Cat. EP0402, ThermoScientific, Waltham, MA USA)
Lines: 390-396.
1.2% agarose gels (Cat. 15510-019, Invitrogen, Grand Island, NY, USA) in 1X TBE. The gels were run at 100 V for around 50 min, and a 100 bp marker (Invitrogen, Cat. 15628019) was used. The products and the marker were stained with GelRed (10,000X), (Cat. 41003, Biotium, Fremont, CA, USA). The fluorescence of the agarose gel was captured using a photodocumenter (ENDURO™ GDS Touch II, Labnet). Subsequently, isoform expression was evaluated using semi-quantitative analysis of the band intensity using the optical densitometry software ImageJ (NIH) [55].
Lines: 402.
Prism (GraphPad ™ Version 8) was used for data analysis and graph plotting.
Reviewer #2 14- Authors checked the normality of distribution of the results by Shapiro-Wilk test before deciding to choose certain ANOVA .
Response: Thank you for pointing out the need to check the normality of the distribution. We have addressed this concern, and the tables of the Shapiro-Wilk test results have been added to the supplementary material. We checked the normality of the distribution of the results using the Shapiro-Wilk test before deciding to use a specific ANOVA. All values followed a normal distribution. We have included this step in the results section.
Please see: Section 2.3 OECs, dBM-MSCs, and uBM-MSCs express GFAPα, GFAPβ, and GFAPδ isoforms. Pages: 4. Lines: 132-145
In the PCR analysis, we examined the GFAP isoforms (α, β, and δ) in OECs, dBM-MSCs, and uBM-MSCs in the PCR analysis. In our analysis, we used the Shapiro-Wilk test to assess the normality of the data. The p-values obtained were above 0.05, indicating that the data did not significantly deviate from a normal distribution. Additionally, we employed ANOVA and Tukey's post hoc tests to determine statistically significant differences between groups. The OECs are characterized by an increased expression of GFAPα (10.96% ± 2.47) with statistically significant differences between groups [F (2,6) = 7.95, p = 0.020], as compared to dBM-MSCs (3.99% ± 1.18; p = 0.049) and uBM-MSCs (2.47% ± 0.44; p = 0.022). In addition, the expression of GFAPβ was also increased (9.82 % ± 0.78) with statistically significant differences between groups [F (2, 6) = 128.1, p< 0.0001] as compared to dBM-MSCs (1.03 % ± 0.16; p< 0.0001) and uBM-MSCs (0.34% ± 0.12; p< 0.0001) (Figure 3 A-E). However, the three groups expressed a similar amount of GFAPδ (OECs, 7.60 % ± 1.94; dBM-MSCs, 8.49 % ± 2.43; uBM-MSCs, 7.54% ± 2.24), and no statically significant differences were found between groups (F (2, 6) = 0.057, p=0.94, Figure 3F).”
Reviewer #2 15- Authors should confirm in methods that "every possible comparison between the study groups was considered" and apply this in results.
Response: As we mentioned above, we have addressed this concern by adding the Tukey test tables in the supplementary material and reorganizing the description of the results to include each comparison.
Reviewer #2 16- Use appropriate abbreviations for minutes, seconds...etc .
Response: Thank you for your observation. We reviewed the manuscript and identified inconsistencies in the abbreviations for minutes and other time units. These have now been corrected to ensure appropriate and consistent usage throughout the text. We appreciate your feedback, which has helped us improve of our work.
Please see: Section of Materials and Methods. Pages: 8-11.
Reviewer #2 17- Every abbreviation in figures should be explained in the figure legend to be self explanatory & stands alone .
and
Reviewer #2 18- Mention "n" in each illustration individually .
Response: Thank you for your insightful suggestion. Following your recommendation, we have incorporated explanations for all abbreviations in the figure legends to ensure they are self-explanatory and can stand alone. Additionally, we have included the "n" value in each illustration individually for clarity.
Please see: Section of Results.
Page 3, Lines: 105-106.
Figure 1. OECs: olfactory ensheathing cells, BM-MSC: Bone marrow-mesenchymal stem cell, dBM-MSC: differentiated BM-MSC, uBM-MSC: undifferentiated BM-MSC. n=3.
Page: 4. Lines: 128-129.
Figure 2. OECs: olfactory ensheathing cells, BM-MSC: Bone marrow-mesenchymal stem cell, dBM-MSC: differentiated BM-MSC, uBM-MSC: undifferentiated BM-MSC. n=3.
Page: 4. Lines: 156-158.
Figure 3. (n=3). OECs: olfactory ensheathing cells, BM-MSC: Bone marrow-mesenchymal stem cell, dBM-MSC: differentiated BM-MSC, uBM-MSC: undifferentiated BM-MSC.
- Discussion
Page: 12. Line: 434.
Supplementary material Figure 1. Cell viability by MTT assay. Data is presented as mean ± SD (n=3),
Reviewer #2 19- Please mention what is the clinical potential of this study and how it can be useful for treating diseases .
Response: Thank you for your thoughtful comment. GFAP is the main biomarker of astrogliosis in neurodegenerative and neurological diseases. Identifying specific GFAP proteoforms could have significant clinical potential by enabling the development of tests for specific biomarkers of neurodegenerative diseases. This would involve recognizing the expression of each isoform associated with particular cellular processes, potentially improving diagnostic precision and advancing targeted therapeutic approaches. We appreciate your feedback, which allowed us to highlight the translational relevance of our study.
Please see: Section of Discusion. Pages: 5-6. Lines: 164-175.
On the one hand, there are novel methods capable of detecting biomarkers in the form of miRNA and mRNA in body fluids including blood and cerebrospinal fluid, as a potential strategy to detect neurodegenerative diseases and inflammatory processes[24,25] , and on the other hand we have that GFAP is the main biomarker of astrogliosis for neuro-degenerative and neurological diseases, thus, identifying specific GFAP proteoforms could result in testing for specific biomarkers of neurodegenerative diseases, by recog-nizing the expression of each isoforms that are characteristic of particular cellular processes [25] for this reason in this study, we were interested in characterizing the expression of GFAPα, GFAPβ, and GFAPδ mRNA by PCR assay in cell culture of uBM-MSCs), dBM-MSCs and OECs obtained from the rat olfactory bulb, as a starting point to establish a relationship between GFAP isoforms and the processes involved in the wound healing process.
- Wang, W. X., Fardo, D. W., Jicha, G. A., & Nelson, P. T. (2017). A Customized Quantitative PCR MicroRNA Panel Provides a Technically Robust Context for Studying Neurodegenerative Disease Biomarkers and Indicates a High Correlation Between Cerebrospinal Fluid and Choroid Plexus MicroRNA Expression. Molecular neurobiology, 54(10), 8191–8202. https://doi.org/10.1007/s12035-016-0316-2.
- Gogishvili, D., Honey, M. I. J., Verberk, I. M. W., Vermunt, L., Hol, E. M., Teunissen, C. E., & Abeln, S. (2025). The GFAP proteoform puzzle: How to advance GFAP as a fluid biomarker in neurological diseases. Journal of neurochemistry, 169(1), e16226. https://doi.org/10.1111/jnc.16226.
Reviewer #2 20- Please mention the limitations of this study and future directions after which.
Response: We appreciate the reviewer’s observation regarding the importance of including references to recent studies and methodologies. In response, we had a section of Limits of the study by incorporating additional references to recent research that explores similar cellular phenomena using advanced techniques. This inclusion highlights opportunities for future work, as our study opens a wide range of perspectives for further investigation. Specifically, we have referenced studies utilizing modern approaches, such as the Scratch Wound Healing Assay, migration assays, and marker-based RT-PCR analyses, to investigate deeply into GFAP isoform expression and its implications for cellular plasticity and regeneration.
Please see: Section of Limitations of the study. Page: 12. Lines: 424-429.
This article recognizes the methodological limitations that future studies could address. Techniques such as the Scratch Wound Healing Assay and migration assays could enhance the current findings [56,57]. These approaches would allow for a more detailed evaluation of GFAP isoform expression and enable their quantitative analysis through PCR, providing deeper insights into their functional roles.

Reviewer 3 Report
Comments and Suggestions for Authors
The authors provided additional details on how the expression of GFAP isoforms specifically contributes to cellular plasticity and regeneration. This can help readers better understand the functional relevance of the findings.
1. I think the authors should include a wound-healing assay or a migration assay to validate the role of GFAP isoforms in cell motility.
2. The authors need to specify the exact composition of the olfactory ensheathing cell-conditioned medium (OECs-CM).
3. The authors should enhance the clarity of their figures, particularly the immunocytochemistry images, by including scale bars and improving resolution.
4. I suggest that the authors investigate and report on the subcellular localization of GFAP isoforms using confocal microscopy.
5. The authors need to briefly discuss the potential therapeutic applications of their findings.
6. I think the authors should expand on the limitations of their study, particularly the absence of in vivo validation.
Author Response
Reviewer #3
The authors provided additional details on how the expression of GFAP isoforms specifically contributes to cellular plasticity and regeneration. This can help readers better understand the functional relevance of the findings.
Reviewer #3.1 I think the authors should include a wound-healing assay or a migration assay to validate the role of GFAP isoforms in cell motility.
Response: We appreciate Reviewer #3's suggestion regarding the inclusion of a wound-healing or migration assay. We recognize that these assays are commonly used to assess cell motility and would provide further insights into the role of GFAP isoforms in this process. However, incorporating such experiments would significantly expand the scope of the current study, which is primarily focused on characterize the expression of GFAPα, GFAPβ, and GFAPδ mRNA isoforms by PCR assay in cell culture rat bone marrow undifferentiated MSCs (uBM-MSCs), glial-differentiated MSCs (dBM-MSCs, exposed to OECs-CM) and OECs obtained from rat olfactory bulb. Our current study focuses on demonstrating OECs, dBM-MSCs, and uBM-MSCs expressing GFAPα, GFAPβ, and GFAPδ isoforms.
We believe that the data presented here provides compelling evidence for demonstrating OECs, dBM-MSCs, and uBM-MSCs expressing GFAPα, GFAPβ, and GFAPδ isoforms within this defined scope. We concur that future studies incorporating wound-healing or migration assays are warranted to further explore the direct effects of GFAP isoforms on cell migration, and we intend to pursue this line of investigation in our future work.
Reviewer #3.2. The authors need to specify the exact composition of the olfactory ensheathing cell-conditioned medium (OECs-CM).
Response: We appreciate Reviewer #3's valuable feedback regarding the need to specify the exact composition of the OECs-CM. We concur that a comprehensive analysis of the conditioned medium would significantly enhance the interpretation of our findings and contribute to the reproducibility of our work. This current study is primarily focused on characterize the expression of GFAPα, GFAPβ, and GFAPδ mRNA isoforms by PCR assay in cell culture rat bone marrow undifferentiated MSCs (uBM-MSCs), glial-differentiated MSCs (dBM-MSCs, exposed to OECs-CM) and OECs obtained from rat olfactory bulb.
We believe the present findings, demonstrating OECs, dBM-MSCs, and uBM-MSCs expressing GFAPα, GFAPβ, and GFAPδ isoforms, make a significant contribution to the understanding of GFAP expression across regeneration promoting cells. We highly value the reviewer's suggestion and have already prioritized experiments to address this critical aspect. Our future work will focus on identifying and quantifying the specific growth factors, cytokines, and other molecules present in the conditioned medium using techniques such as LC-MS/MS and multiplex assays. This detailed analysis will be crucial for elucidating the precise mechanisms by which the OECs-CM exerts its effects and will be a key focus of our future research efforts. We appreciate the reviewer's feedback, and this point will be addressed in forthcoming work.
Reviewer #3.3- The authors should enhance the clarity of their figures, particularly the immunocytochemistry images, by including scale bars and improving resolution.
Response: We thank Reviewer #3 for this constructive comment. We concur that the addition of scale bars and improved resolution would be beneficial for the interpretation of our immunocytochemistry data. We have now added scale bars to each image and have also provided zoomed-in views of the regions of interest. We believe this significantly enhances the visualization of the data and clarifies the points we are making. We are grateful to the reviewer for this helpful suggestion.
Please see: Section 2. Results: Figure 1, Page: 3,. Figure 2, Page: 4.
Figure 1. OECs, dBM-MSCs, and uBM-MSCs phenotypic characterization. (A) OECs showed elongated spindle-shape and small cell body morphology with bipolar (white arrow) or multipolar (yellow arrow) ramifications and positive expression to p75 and GFAP. (B) dBM-MSCs mainly showed elongated spindle shape and small cell body morphology (white arrow) with positive expression to p75 and GFAP. These data suggest that dBM-MSCs had a Schwann-like phenotype after exposure to OECs-CM. (C) uBM-MSCs showed a fibroblast-like morphology (white arrow) and positive expression of CD90 and GFAP. OECs: olfactory ensheathing cells, BM-MSC: Bone marrow-mesenchymal stem cell, dBM-MSC: differentiated BM-MSC, uBM-MSC: undifferentiated BM-MSC. n=3. Bar = 100 µm.
Figure 2. Characterization of GFAP distribution in OECs, dBM-MSCs, and uBM-MSCs. Three groups showed diffuse GFAP throughout the cytoplasm. (A) OECs present a mostly spindle-like morphology with long extensions and diffuse GFAP (asterisk). (B) dBM-MSCs expressed GFAP in cell types with spindle-like morphology. (C) uBM-MSCs expressed GFAP in mostly cells with morphology similar to fibroblasts (flattened fibroblast) (C). OECs: olfactory ensheathing cells, BM-MSC: Bone marrow-mesenchymal stem cell, dBM-MSC: differentiated BM-MSC, uBM-MSC: undifferentiated BM-MSC. n=3. Bar = 50 µm.
Reviewer #3.4 I suggest that the authors investigate and report on the subcellular localization of GFAP isoforms using confocal microscopy.
Response: We appreciate Reviewer #3's keen comment regarding the subcellular localization of GFAP isoforms. We acknowledge that confocal microscopy would provide valuable insights into the precise localization of these isoforms and their potential functional implications, such as migration, plasticity and neural repair and regeneration. While the current study focuses on characterize the expression of GFAPα, GFAPβ, and GFAPδ mRNA isoforms by PCR assay in uBM-MSCs, dBM-MSCs, and OECs, we recognize the importance of determining their subcellular localization for a complete understanding of their role. We have already included confocal microscopy analysis of GFAP isoform localization in our plans for future investigations. We are grateful for the reviewer's valuable suggestions, that reinforce the importance of this line of inquiry.
Reviewer #3.5- The authors need to briefly discuss the potential therapeutic applications of their findings.
Response: We appreciate Reviewer #3's feedback regarding the therapeutic implications of our work. We have now expanded the Discussion section to include a brief discussion of the potential therapeutic applications of our findings.
Please see: Section of Discussion. Pages:5-6 Lines: 164-175.
On the one hand, there are novel methods capable of detecting biomarkers in the form of miRNA and mRNA in body fluids including blood and cerebrospinal fluid, as a potential strategy to detect neurodegenerative diseases and inflammatory processes[24,25] , and on the other hand we have that GFAP is the main biomarker of astrogliosis for neuro-degenerative and neurological diseases, thus, identifying specific GFAP proteoforms could result in testing for specific biomarkers of neurodegenerative diseases, by recognizing the expression of each isoforms that are characteristic of particular cellular processes [25] for this reason in this study, we were interested in characterizing the expression of GFAPα, GFAPβ, and GFAPδ mRNA by PCR assay in cell culture of uBM-MSCs), dBM-MSCs and OECs obtained from the rat olfactory bulb, as a starting point to establish a relationship between GFAP isoforms and the processes involved in the wound healing process.
- Gogishvili, D., Honey, M. I. J., Verberk, I. M. W., Vermunt, L., Hol, E. M., Teunissen, C. E., & Abeln, S. (2025). The GFAP proteoform puzzle: How to advance GFAP as a fluid biomarker in neurological diseases. Journal of neurochemistry, 169(1), e16226. https://doi.org/10.1111/jnc.16226.
Reviewer #3.6- I think the authors should expand on the limitations of their study, particularly the absence of in vivo validation.
Response: We thank Reviewer #3 for this constructive comment. In response, we had a section of Limits of the study by incorporating additional references to recent research that explores similar cellular phenomena using advanced techniques. This inclusion highlights opportunities for future work, as our study opens a wide range of perspectives for further investigation. Specifically, we have referenced studies utilizing modern approaches, such as the Scratch Wound Healing Assay, migration assays, and marker-based RT-PCR analyses, to investigate deeply into GFAP isoform expression and its implications for cellular plasticity and regeneration.
Please see: Section of Limitations of the study. Page: 12. Lines: 424-429.
This article recognizes the methodological limitations that future studies could address. Techniques such as the Scratch Wound Healing Assay and migration assays could enhance the current findings [56,57]. These approaches would allow for a more detailed evaluation of GFAP isoform expression and enable their quantitative analysis through PCR, providing deeper insights into their functional roles.

Round 2
Reviewer 1 Report
Comments and Suggestions for Authors
Good
Reviewer 2 Report
Comments and Suggestions for Authors
The revised version of the paper titled ( GFAPβ and GFAPδ isoforms expression in mesenchymal stem cells, MSCs differentiated towards Schwann-like and Olfactory Ensheathing Cells) submitted to CIMB by Authors Nidia Jannette Carrillo-González et al. was improved compared to the original submission and authors performed all the required revisions and hence I now can recommend this paper for publication in CIMB
Thanks
Reviewer 3 Report
Comments and Suggestions for Authors
No more comments